# Identifying Complex DNA Contamination in Pig-Footed Bandicoots Helps to Clarify an Anomalous Ecological Transition

**Matthew J. Phillips** *  , **Manuela Cascini** and **Mélina Celik**

School of Biology and Environmental Science, Queensland University of Technology, 2 George Street, Brisbane, QLD 4000, Australia; manuelacascini@gmail.com (M.C.); melina.celik@gmail.com (M.C.)
* Correspondence: m9.phillips@qut.edu.au

**Abstract:** Our understanding of the biology of the extinct pig-footed bandicoots (*Chaeropus*) has been substantially revised over the past two decades by both molecular and morphological research. Resolving the systematic and temporal contexts of *Chaeropus* evolution has relied heavily on sequencing DNA from century-old specimens. We have used sliding window BLASTs and phylogeny reconstruction, as well as cumulative likelihood and apomorphy distributions, to identify contamination in sequences from both species of pig-footed bandicoot. The sources of non-target DNA that were identified range from other bandicoot species to a bird—emphasizing the importance of sequence authentication for historical museum specimens, as has become standard for ancient DNA studies. Upon excluding the putatively contaminated fragments, *Chaeropus* was resolved as the sister to all other bandicoots (Peramelidae), to the exclusion of bilbies (*Macrotis*). The estimated divergence time between the two *Chaeropus* species also decreases in better agreement with the fossil record. This study provides evolutionary context for testing hypotheses on the ecological transition of pig-footed bandicoots from semi-fossorial omnivores towards cursorial grazers, which in turn may represent the only breach of deeply conserved ecospace partitioning between modern Australo-Papuan marsupial orders.

**Keywords:** Peramelemorphia; marsupials; phylogeny; ecospace; DNA authentication

## 1. Introduction

Bandicoots and bilbies (Peramelemorphia) are unusual among living marsupials in possessing a rudimentary chorio-allantoic placenta with umbilicus [1], and a robust patella [2] (also *Notoryctes* and Caenolestidae [3,4]). The two extant peramelemorphian families include the lone surviving bilby (*Macrotis*: Thylacomyidae) and ~22 species of bandicoots (Peramelidae). To varying extents, all are semi-fossorial omnivores, digging and foraging terrestrially for invertebrates, bulbs, fungi and fruit. Pig-footed bandicoots (*Chaeropus*) appear to have evolved into novel ecospace for peramelemorphians. They are proposed to have been cursorial grazers [5,6], characterized by gracile, two-toed forelimbs and higher-crowned teeth. Their relationships and temporal divergence from other peramelemorphians have been contentious. Morphological assessments [7–10] have been tentative and may be compromised by Oligo-Miocene fossil taxa being drawn (apparently artefactually) towards the plesiomorphic or secondarily "primitive" [11] and primarily New Guinean Peroryctinae (among which we include *Peroryctes*, *Echymipera*, *Rhynchomeles* and *Microperoryctes*). Nevertheless, these studies tended to toggle between placing *Chaeropus* as sister to Thylacomyidae (*Macrotis*) or close to Peramelinae (*Isoodon* and *Perameles*), within Peramelidae.

Analyses of DNA sequences have further inflated the uncertainty surrounding *Chaeropus* affinities (see Figure 1). The study by Westerman et al. [12] was the first molecular phylogenetic study to include *Chaeropus*. They employed mitochondrial 12S rRNA

sequences and placed *Chaeropus* as the sister to all other extant bandicoots and bilbies. Meredith et al. [13] published a partial, nuclear *RAG1* sequence from *Chaeropus*, which they favoured grouping with Peramelidae to the exclusion of *Macrotis*. Additional, partial 16S rRNA and *Cytb* mtDNA sequences were published in association with Westerman et al. [14], although only the 16S sequence was included for analysis alongside the available 12S and *RAG1* sequences. That study strengthened support for *Chaeropus* falling outside all other peramelemorphians. May-Collado et al. [15] utilized all sequences available at the time to construct their marsupial supermatrix tree, in which *Chaeropus* was sister to the peroryctine bandicoots. Subsequently, Travouillon and Phillips [16] combined the mtDNA sequences with morphological data (including for fossil taxa) and placed *Chaeropus* as sister to *Macrotis* or outside all extant peramelemorphians (as did Kear et al. [17] and Beck et al. [18]). Travouillon and Phillips [16] cautioned the use of the *Chaeropus RAG1* sequence, because its phylogenetic signal is largely confined to ambiguous sites and, unusually, the inferred substitutions along this lineage are dominated by transversions over transitions. *Cytb* provided the most remarkable result, with Upham et al. [19] nesting the *Chaeropus yirratji* sequence within a different marsupial order (Dasyuromorphia), among the dunnarts (*Sminthopsis*).

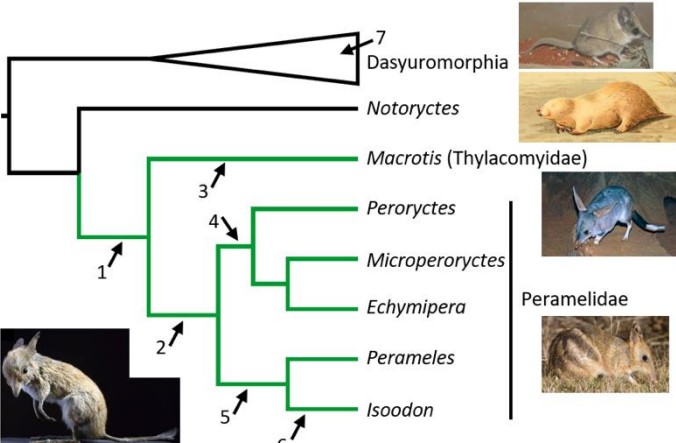

**Figure 1.** Peramelemorphia phylogeny (in green) for all extant genera, with dasyuomorphian and *Notoryctes* outgroups. The potentially extinct *Rhynchomeles* may fall within *Echymipera* [16]. Placements of *Chaeropus* in molecular and combined molecular-morphological studies; 1. Westerman et al. [12,14], Kear et al. [17], Travouillon and Phillips [16], Beck et al. [18], 2. Meredith et al. [13], Travouillon et al. [20], 3. Travouillon and Phillips [16], 4. May-Collado et al. [15], 5,6. Travouillon et al. [6], 7. Upham et al. [19]. Note that alternative analyses in some studies favoured differing placements. Images: Left; *Chaeropus yirratji* (Muséum national d'histoire naturelle, Paris), Right, from the top; *Sminthopsis crassicaudata* (A Couch), *Notoryctes typhlops* (R Lydekker), *Macrotis lagotis* (B Dupont), *Perameles gunnii* (JJ Harrison).

　　　Travouillon et al. [6] provided the most comprehensive phylogenetic examination yet, for the affinities of *Chaeropus*. They obtained new mtDNA (12S/16S rRNA and *Cytb*) sequences and scored craniodental morphological characters across extant and fossil bandicoots. Both datasets placed *Chaeropus* in a clade with *Perameles* and *Isoodon*; the DNA favoured a closer relationship with *Isoodon* and the morphology tended to favour a closer relationship with *Perameles*. Travouillon et al.'s [6] study is also important for lending molecular and morphological support for splitting *Chaeropus* into two species, *C. ecaudatus* with a semi-arid distribution and *C. yirratji* with a more arid distribution. Several morphological characters supported this taxonomic distinction, including maxillary fenestrae in *C. ecaudatus*, and larger metaconules in *C. yirratji*, lending additional blades for processing plant material. Travouillon et al.'s [6] molecular divergence estimate for these two recently extinct species of 8.6 (95% CI: 3.2–13.4) Mya is surprisingly old in view of the 2.92–2.47 Mya

age of the more plesiomorphic stem fossil taxon, *C. baynesi* from the Fisherman's Cliff Local Fauna [21,22].

It is important to examine the authenticity of published *Chaeropus* DNA sequences in light of substantial, apparent phylogenetic incongruence between genes and the potential temporal discrepancy between the divergences of stem and crown taxa. Enormous strides have been made in the authentication of DNA sequences in the fields of ancient DNA and forensics, with protocols ranging from replicating sequences in different labs (e.g., [23]) to profiling patterns of DNA fragment length and damage [24]. These methods tend to be upstream in the experimental and analytical process. Once sequences are published, however, authentication typically requires phylogenetic methods, such as analysis of evolutionary rates [25], similarity measures [26] and topological agreement [27].

To authenticate published *Chaeropus* sequences, we employed phylogenetic authentication methods within a sliding window framework (e.g., [28,29]). This approach allowed smaller non-target DNA fragments to be identified within longer sequences. Phylogenetic analyses of the remaining set of more confidently attributed sequences were undertaken to clarify the affinities and temporal divergence of *Chaeropus*, which in turn lend a novel context for understanding their ecological transition from semi-fossorial omnivores towards cursorial grazers.

## 2. Materials and Methods

### 2.1. Sequence Authentication of GenBank DNA Accessions

Nine mtDNA sequences attributed to *Chaeropus* have been published to date. These include Sanger sequenced 12S rRNA (AF131247), 16S rRNA (JF706364) and *Cytb* (JF718363) from *Chaeropus yirratji*, published by Westerman et al. [12,14], and Illumina sequenced 12S rRNA (MK359293, MK359294), 16S rRNA (MK359295, MK359296) and *Cytb* (MK359297, MK359298) from *Chaeropus ecaudatus*, published by Travouillon et al. [6]. The two *C. ecaudatus* sequences for each of these genes are identical. Hence, 12S rRNA, 16S rRNA and *Cytb* are effectively available for two taxonomic units, which are the two species. These published *Chaeropus* mtDNA sequences were initially BLASTed against other mammal sequences in GenBank, using The NCBI's discontinuous megablast [30]. These BLASTs were undertaken for windows of 150 bp that were slid in steps of 75 bp until the end of each sequence. We tried several window widths and 150 bp was an acceptable compromise, as it was long enough to be informative on potential contamination, whilst not being too long to isolate the position of contaminated fragments. Sequences that did not closely match any mammals were subsequently BLASTed without taxonomic constraint.

Sliding window analysis was extended to maximum parsimony (MP) phylogeny reconstruction, with *Chaeropus* allowed to float on a backbone constraint tree of 190 other marsupials, allowing fine-grained assessment of contamination. MP Bootstrap trees were obtained in PAUP 4.0b10 [31] from 250 heuristic search pseudoreplicates for windows of 300 bp that were slid in steps of 100 bp until the end of each sequence. These longer windows were necessary to improve phylogenetic resolution. Subsequent parsimony apomorphy reconstructions in PAUP employed the MP tree for the concatenated mtDNA with the backbone constraint and monophyly enforced for *C. ecaudatus* and *C. yirratji*. In these apomorphy reconstructions, *Chaeropus* was sister to Peramelidae, which is in agreement with the maximum likelihood and Bayesian inference trees (see below).

Alternative phylogenetic placements for *Chaeropus* that were identified in the MP bootstrap sliding window analyses were subsequently compared for cumulative site likelihood along each of the three genes. Site likelihoods were inferred in IQ-TREE [32] for each of the alternative placements, using the partition schemes, substitution modelling, and the $Mt_{192}$ constraint tree that are outlined below.

### 2.2. Phylogenetic Inference

Two primary data sets were employed to infer the phylogenetic relationships of *Chaeropus*. The first of these, the 3149 bp $Mt_{192}$ dataset, included only the three mtDNA

genes that are available for *Chaeropus* (12S and 16S rRNA, *Cytb*). In addition to the two *Chaeropus* species, these data include 190 other taxa that cover all modern marsupial families and all Australo-Papuan marsupial genera except for two monotypic ringtail possum genera (*Hemibelideus* and *Petropseudes*), for which mtDNA sequences were unavailable.

To enhance the potential for identifying contamination with the short (300 bp) sliding windows, we constructed a constraint tree for the 190 non-*Chaeropus* taxa. The first step was inferring an unconstrained ML tree in IQ-TREE for the $Mt_{192}$ data matrix, without *Chaeropus*. This 12/16S-*Cytb* tree (Figure S1) provides close agreement with the most comprehensively gene-sampled and well-resolved genome-scale marsupial tree [33] and also with more densely taxon-sampled nuclear-mitogenomic supermatrix trees (e.g., [16,34–36]). Minor differences from expected relationships were corrected in a second ML analysis with topological constraints conforming to Duchêne et al. [33] and enforcing monophyly for each of Peroryctinae, *Perameles*, *Pseudantechinus*, and *Sarcophilus-Dasyurus*. This produced the final $Mt_{192}$ constraint tree on which *Chaeropus* placement could float for the sliding window and cumulative site likelihood analyses of the mtDNA (see Figure S2). For the present work, it is important that this constraint tree conforms to the mitochondrial tree, not the marsupial species tree. As such, the swamp wallaby (*Wallabia bicolor*), which has an introgressed mitogenome [37,38], was constrained to be sister to the other large kangaroos and wallabies (*Macropus*, *Osphranter*, *Notamacropus*).

The second dataset ($MtNuc_{26}$) is modified from Travouillon and Phillips [16] by adding the *C. ecaudatus* sequences alongside *C. yirratji*, 16 extant peramelemorphian species, and nine outgroup marsupials. The three mtDNA genes are included and supplemented with five nuclear genes (*BRCA1*, *IRBP*, *RAG1*, *ApoB* and *vWF*) that have been broadly sampled across extant bandicoots. The additional gene sampling for the 9314 bp $MtNuc_{26}$ is intended to clarify relationships and divergences among extant bandicoots and in turn, to improve inference of the relationships and timescale of *Chaeropus* evolution.

The two data matrices ($Mt_{192}$ and $MtNuc_{26}$) were manually aligned in Se-Al 2.0a [39]. Model partitions followed Travouillon and Phillips [16] for $MtNuc_{26}$. The mtDNA was partitioned into rRNA stems, rRNA loops and the three *Cytb* protein-codon positions, while the five nuclear gene sequences were concatenated and partitioned into their three protein-coding positions. Given the emphasis on cumulative site likelihoods across genes with the $Mt_{192}$ data, for those analyses, the rRNA data were instead sequentially partitioned as 12S and 16S rRNA. Substitution models for each partition (Table S1) were assigned in accordance with ModelFinder results obtained with IQ-TREE v1.6.10 [32]. Maximum likelihood analyses were performed in IQ-TREE. Corrected AIC (AICc) favoured estimating the branch lengths independently across partitions (-sp option) for $MtNuc_{26}$ and with branch length multipliers (i.e., proportional across partitions, -spp option) for $Mt_{192}$.

Bayesian phylogenetic inference of $MtNuc_{26}$ was carried out with MrBayes 3.2.7 [40]. Two independent runs each included three Markov Chain Monte Carlo (MCMC) chains for five million generations. The same partitions and substitution models (or the next most general available in MrBayes) were used as described above for ML. Models were unlinked across all partitions for the substitution matrix, (empirical) state frequencies, proportions of invariant sites and the shape parameter of the rates-across-sites gamma distribution. Branch lengths were unlinked between the nuclear and mtDNA data, but they were proportionally scaled across partitions within each of these genomes. Trees were sampled every 5000 generations, with the first 25% discarded as burn-in. Clade frequencies across the two independent runs reached convergence (clade frequency standard deviations < 0.01) and estimated sample sizes for likelihood, prior and substitution parameter estimates were all above 200 (Tracer v1.7.1 [41]).

Mitochondrial protein 3rd codon sites are particularly susceptible to a combination of phylogenetic signal erosion and nucleotide compositional heterogeneity, which can mislead phylogenetic inference, including for marsupials [42,43]. Hence, we ran the primary maximum likelihood and Bayesian inference phylogenetic analyses ($MtNuc_{26}$) with all sites standard (NT) coded and alternatively, with *Cytb* 3rd codon positions RY-coded (A,G $\rightarrow$ R;

C,T → Y). Having identified potentially contaminated regions in several of the *Chaeropus* sequences, our primary phylogenetic analyses excluded those regions.

### 2.3. Molecular Dating

Divergence times were estimated in BEAST v.1.8.1 [44] using the uncorrelated relaxed clock model with lognormally distributed branch rates [45]. The MtNuc$_{26}$ data matrix was partitioned as described for the MrBayes phylogenetic analyses and was run alternatively with standard NT-coding and with RY-coding for the *Cytb* 3rd codon positions. Eight fossil-based prior age distributions were modified from Travouillon and Phillips [16] to provide node calibration (see Table S2).

Each BEAST analysis was run for 40,000,000 MCMC generations, with the chain sampled every 5000th generation, following a burn-in of 4,000,000 generations. This resulted in estimated sample size values >100 (estimated in Tracer v1.71) for −ln*L*, tree and substitution parameters, and importantly, for all node heights. However, ESS values for the prior and consequently for the posterior were low (between 20–50). Therefore, we ran two additional, independent 15,000,000 generation runs for verification. These gave posterior node heights that were essentially identical to the primary analyses.

## 3. Results

### 3.1. Authentication of GenBank DNA Sequences

The Mt$_{192}$ data matrix includes 12S/16S rRNA and *Cytb* accessions for 192 marsupials, including both *Chaeropus ecaudatus* and *C. yirratji*. ML analyses of these data fail to recover *Chaeropus* as monophyletic. *C. ecaudatus* is sister to Peramelidae and *C. yirratji* is sister to all bandicoots and bilbies (Figure S1). To identify potential contamination, a sliding window approach with BLASTs and MP bootstrap was employed to examine phylogenetic signal variation along individual sequence accessions.

As a starting point for authentication, three systematic expectations for pig-footed bandicoots can be confidently derived from morphology (see [6]): *Chaeropus* is (1) unambiguously peramelemorphian, (2) a distinct genus, separate from other lineages, and (3) monophyletic. Thus, in principle, authentic *Chaeropus* sequences should BLAST and phylogenetically group with peramelemorphians, but not with far closer affinity to any particular peramelemorphian species or genus to the exclusion of others (except for congeneric *Chaeropus* accessions). To gauge how these expectations might fare with real data, which involves biases from sources such as base compositional heterogeneity and stochastic artefacts with short windows, we first applied our sliding window BLAST and MP bootstrap approach to the bilby (*Macrotis lagotis*, AJ639871). *Macrotis* is valuable for guiding prior expectations for *Chaeropus* sequences, because both are taxonomically isolated lineages that are most often thought to have diverged along the stem lineage leading to peramelid bandicoots.

Using *Macrotis* as a control shows that authentic peramelemorphian sequences diverging from the peramelid stem lineage will not necessarily BLAST or phylogenetically place closest to other bandicoots. Among the *Macrotis* rRNA and *Cytb* BLAST windows, respectively, 23% and 64% of top hits were outside of Peramelemorphia (Table S3). Similarly, 25% and 67%, respectively, of rRNA and *Cytb* sliding window MP bootstraps favoured *Macrotis* placements outside of Peramelemorphia (Table S4). We found better success in circumscribing expectations for authentic sequences by using two metrics. The first metric we refer to as the "identity ratio", which is the specificity of the top BLAST hit. Where the top BLAST hit (percentage identity) with any taxon is I$_A$ and the next highest BLAST hit to a peramelemorphian is I$_B$:

$$\text{Identity ratio} = (1 - I_A)/(1 - I_B) \tag{1}$$

Since congeneric taxa will often be highly similar and thus may mask the specificity of potential contamination, if I$_A$ is a peramelemorphian, then I$_B$ will be taken as the highest hit for another peramelemorphian genus.

The second metric we refer to as "anomalous MP bootstrap support", which is the highest bootstrap support for any placement of the focal taxon that is incongruent with prior expectations. For *Chaeropus*, this will be whichever is higher, the bootstrap support for its exclusion from other peramelemorphians and *Notoryctes* or the bootstrap support for being a shallow-level sister group with or within another genus (peramelemorphian or not). We do not assume that *Chaeropus* is not sister to another bandicoot genus, but it is unlikely to be so close to (or within) another genus that a 300 bp window would provide high (e.g., ≥95% bootstrap support).

Plotting the identity ratio versus anomalous MP bootstrap support for each sliding window (Figure 2) shows a close match between the distributions for *Macrotis* and *C. ecaudatus*. The area bounded by 95% of these *Macrotis* and *C. ecaudatus* sliding window data points sets an expectation for authentic *Chaeropus* sequences. It may even be somewhat conservative, since the *C. ecaudatus* cluster tends to fall within the lower half (17–58%) of anomalous MP bootstrap support values. It is also notable that the MP bootstrap results for the two outlier *Macrotis* BLAST windows still do not reject peramelemorphian affinities at $p = 0.05$, given the anomalous MP bootstrap support is 94% (both of the 150 bp BLAST windows are covered by the same 300 bp MP window in the centre of 16S rRNA). Moreover, that anomalous support is not primarily linked to a particular taxon (the highest genus-level affinity is *Petaurus*, at 7%). Instead, the anomaly relates to these two overlapping BLAST windows being an apomorphic sequence in peramelids, leaving the *Macrotis* sequence plesiomorphically similar to numerous non-peramelemorphians. These considerations lessen concern for the *Macrotis* windows being non-target DNA.

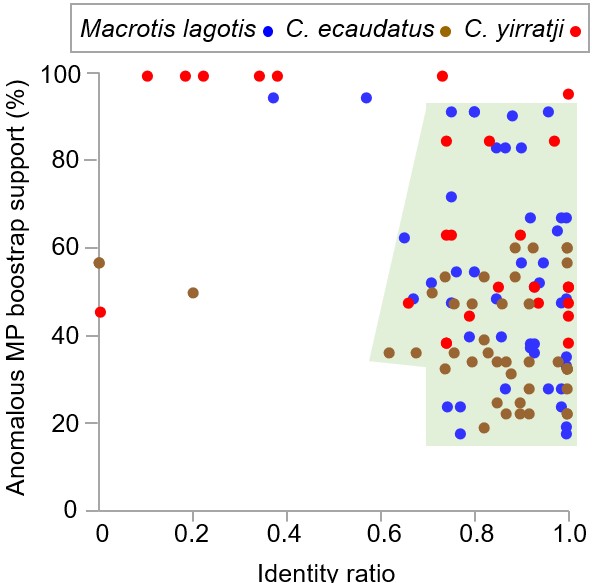

**Figure 2.** Scatterplot of the identity ratio from 150 bp BLAST windows versus highest anomalous MP bootstrap support among 300 bp MP windows that fully include that BLAST window, for *Macrotis lagotis* (blue), *C. ecaudatus* (brown) and *C. yirratji* (red). All BLAST windows are included from 12S/16S rRNA and *Cytb*. In the MP bootstrap analyses, the *Cytb* 3rd codon positions were RY coded. The shaded area covers 95% of *Macrotis* and 95% of *C. ecaudatus* windows. Identity ratio is a metric for the specificity of the identity to the top hit ($I_A$) relative to the identity for the next most similar peramelemorphian genus ($I_B$). Identity ratio = $(1 - I_A)/(1 - I_B)$. Anomalous MP bootstrap support is whichever is the higher discrepancy with prior expectations, the support for the focal taxon either being excluded from other peramelemorphians and *Notoryctes* or being a shallow-level sister group with or within another genus. Scores on these metrics for each window are provided in Tables S3 and S4.

Thus, by considering *Macrotis* as a control, we are able to preliminarily define *Chaeropus* sliding window sequences as either likely authentic, likely non-target DNA or potentially non-target DNA. Likely non-target DNA includes windows corresponding to data points in Figure 2 that fall outside the shaded 95% distribution for BLAST identity ratio versus highest anomalous MP support and have an identity ratio < 0.30 (lowest for *Macrotis* is 0.37) or anomalous MP bootstrap ≥ 95%. Potential non-target DNA includes windows for which two or more of the following conditions are met (see Table S3): (1) the other *Chaeropus* species is not the top hit, (2) the identity ratio is below 0.7, (3) the top hit is outside of Peramelemorphia, (4) both windows immediately before and after are likely or potential non-target DNA or (5) a highly unusual sequence is included that does not closely match any accession.

### 3.1.1. Chaeropus Ecaudatus Sequence Authenticity

The *Chaeropus ecaudatus* 12S rRNA sequences (MK359293, MK359294) sequences match our BLAST and MP bootstrap authenticity expectations for all windows (Tables S3 and S4) and fall within the shaded 95% distribution for BLAST identity ratio versus highest anomalous MP support (Figure 2). Top hits for each of the (150 bp) BLAST windows were bandicoots, closely followed by other bandicoots, such that all identity ratios were high (0.74–1.00). The corresponding 300 bp window MP bootstraps all favoured *Chaeropus* grouping with or within Peramelemorphia, without strong affinity to any particular genus or species.

All of the 300 bp sliding window MP bootstrap analyses for *C. ecaudatus* 16S rRNA (MK359295, MK359296) favour peramelemorphian affinities, without strong support for placements with any particular taxon (Table S4). However, the finer-scaled BLAST (Table S3) and log likelihood (ln*L*, Figure 3D) accumulation results identify two sections that deserve further consideration. There is rapid fluctuation in ln*L* advantage across the first 100 bp for placements within Peramelidae (Figure 3D). Moreover, much of this segment is difficult to align, it shows no closer similarity to marsupials than to placentals (especially otariids) and it is not clear how the first two 16S stem-loop structures from this 5′ end would form. The alignment and authenticity of the MK359295/MK359296 16S sequences are more assured from base position 99, until a stretch of 49 ambiguous "N" nucleotides that is closely followed by a 392 bp fragment (pos 710–1101) that is almost identical to *Isoodon macrourus* (391/392–only a single transition apart). Next most similar is *Isoodon obesulus* (382/392). Such extensive convergence upon *Isoodon* and *I. macrourus,* in particular, is implausible, especially as the variation derives largely from the less functionally constrained 16S rRNA "loop" sites. These windows in the middle of 16S rRNA are the *C. ecaudatus* outliers in Figure 2, with identity ratios of 0.00 and 0.20. Comparison with the Westerman et al. [14] 16S *C. yirratji* sequence supports the authenticity of MK359295/MK359296 from position 1104 onwards.

Examining the authenticity of the *Cytb* sequences is complex, because their rapid evolution facilitates biases (such as nucleotide compositional biases) that can mask true (inherited) phylogenetic similarity. This may help to explain why most *C. ecaudatus Cytb* (MK359297, MK359298) windows BLAST outside of Peramelemorphia (Table S3), similar to *Macrotis*. This hypothesis is consistent with the average anomalous BP bootstrap support across the *Cytb* windows being reduced from 55% to 35% by RY coding the *Cytb* 3rd codon positions. All of the *C. ecaudatus Cytb* windows fall within the shaded 95% distribution for BLAST identity ratio versus highest anomalous MP support (Figure 2) and we find no basis for rejecting their authenticity.

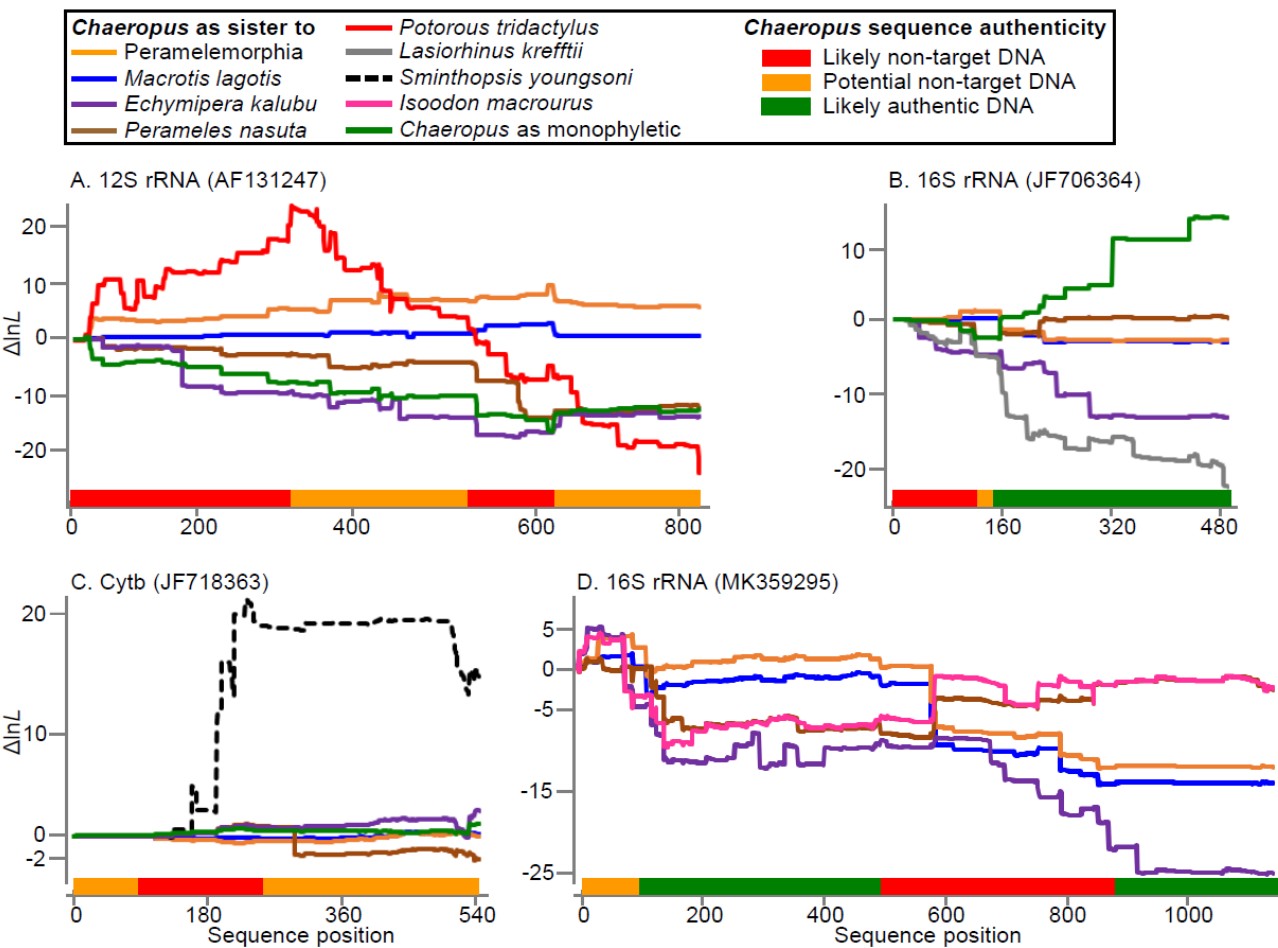

**Figure 3.** Cumulative ln*L* differences along gene sequences, for alternative sister group relationships for the published *Chaeropus yirratji* sequences, (**A**) 12S rRNA (AF131247), (**B**) 16S rRNA (JF706364), (**C**) *Cytb* (JF718363) and the published *Chaeropus ecaudatus* sequence, (**D**) 16S rRNA (MK359295 and MK359296 are identical). In each case, the null (zero Δln*L*) is for the focal *Chaeropus* sequence as sister to Peramelidae. Inferences of the authenticity of these sequences are indicated above the x-axes, as likely non-target DNA (red), potential non-target DNA (orange) or likely authentic DNA (green), based primarily on sliding window BLAST and MP bootstrap with densely sampled marsupial alignments (see Tables S3 and S4).

### 3.1.2. *Chaeropus yirratji* Sequence Authenticity

*Chaeropus yirratji* 12S rRNA (AF131247) appears to be a chimera of several marsupial sequences, including a potoroo (*Potorous tridactylus*), a bilby (*Macrotis lagotis*) and other bandicoots (potentially including *Chaeropus*), and an unidentified fragment. BLAST and MP bootstrap (Tables S3 and S4) show the *Potorous* contamination covers the first third of the sequence and one of the middle windows matches 100% to a bilby sequence. Each of those windows are outliers in Figure 2, either with an identity ratio of 0.0 or anomalous MP bootstrap support of 100%. Cumulative likelihood variation traces ln*L* support along the gene sequences for alternative *Chaeropus* placements on the 192-taxon tree (relative to ln*L* for the *Chaeropus* placement as sister to Peramelidae that was favoured on the full concatenated dataset). The clearest anomaly for 12S rRNA is support for *Potorous* affinities over the first third of the sequence (Figure 3A, red line). Authentic *C. yirratji* sequence is not rejected for the last third of the 12S rRNA accession, but that segment is substantially more similar to other bandicoots than to *C. ecaudatus*. This in itself does not identify which of these *Chaeropus* sequences is artefactual. However, there is otherwise no hint of non-target DNA in the *C. ecaudatus* 12S rRNA sequences (MK359293/359294). In contrast, those final

two sliding window BLASTs for *C. yirratji* 12S (AF131247) include a 31 bp segment that does not closely match any GenBank sequence. Therefore, we suggest that the cautious approach of excluding all of AF131247 is currently expedient.

16S rRNA (JF706364) is the only published sequence for *C. yirratji* with all windows falling within the shaded 95% distribution for BLAST identity ratio versus highest anomalous MP support (Figure 2). It also exhibits the expected close similarity and sister grouping with *C. ecaudatus*, at least for the last two thirds of the sequence (Figure 3B, green line, also Table S3). JF706364 matches 98.8% to *C. ecaudatus* (MK359295/359296) from base 174 onwards, and the next most similar sequences are other bandicoots with several percent lower identity (e.g., *Isoodon obesulus*, 94.9%). This congeneric similarity (and monophyly) shows what could also have been expected from the 12S rRNA and *Cytb* sequences had the accessions from both *Chaeropus* species been authentic.

The first third of the *C. yirratji* 16S rRNA (JF706364) sequence is anomalous. The first 150 bp sliding window BLAST most closely matches the wombat, *Lasiorhinus krefftii* from another order (Diprotodontia) at 98.0% identity. This alone is not necessarily cause for concern, because there is only a small uptick in cumulative ln*L* for this wombat affinity (Figure 3B, grey line) and BLAST matches to bandicoots are not far behind (e.g., *Perameles nasuta* at 97.3%). However, two results raise concern. These are (1) the similarity to wombats is specific to *Lasiorhinus krefftii* (98.0%) but not its close relative *Vombatus ursinus* (92.7% identity) and (2) the first third of 16S is dominated by autapomorphies along the lineage leading to the JF706364 sequence (Figure S3). Alternative explanations could include miscalled bases on an unclear electrophoretogram or that this fragment of sequence has rapidly diverged in *Vombatus* and is contaminated by another bandicoot in the other *Chaeropus* (*ecaudatus*) sequence.

We can only be confident in the JF706364 *C. yirratji* sequence from position 174 onwards and recommend excluding at least the first 130 bp as potentially non-target DNA until confirmed authentic. The intervening sequence (130–174 bp) is identical to *C. ecaudatus*. Therefore, clarifying the authenticity of those 16S sites in the *C. ecaudatus* sequence (see above) could lend veracity for *C. yirratji*. Unfortunately, this segment of 16S does not stem-pair with sites in the remainder of the 16S sequence, precluding another avenue for verification.

The *C. yirratji Cytb* sequence (JF718363) does not appear to be authentic. Most windows fall outside the shaded 95% distribution for BLAST identity ratio versus highest anomalous MP support (Figure 2). Moreover, no similarity (Table S3) or likelihood support for grouping with *C. ecaudatus* emerges along this partial *Cytb* sequence (Figure 3C, green line). All MP bootstrap windows favour placements outside Peramelemorphia (Table S4). The first two of those have 100% and 95% bootstrap support for JF718363 grouping with two dunnarts (*Sminthopsis youngsoni* and *S. ooldea*) within Dasyuromorphia. This coincides with a strong ln*L* signal (Figure 3C, dashed line). Relevant BLAST windows (76–225 bp and 151–300 bp) provide <80% identity to the other bandicoots, but have 96–98% identity to the crested bellbird (*Oreoica gutturalis*) and the above-noted dunnarts. The last two JF718363 BLAST windows provide curious results. They BLAST closest to another dasyurid genus, *Pseudantechinus* (though only ~90%). For the MP bootstrap window that covers those BLASTs, JF718363 falls outside of all other peramelemorphians and marsupial moles, and ML support plateaus then falls for the dasyurid (*Sminthopsis*) in Figure 3C. Taken together, these results for the last two JF718363 BLAST windows might be explained by a chimera of dasyurid and bandicoot sequences.

### 3.2. Phylogenetic Affinities of Chaeropus

To reconstruct the phylogenetic placement of *Chaeropus*, we excluded gene sequence fragments that could not be authenticated and that we considered to be contaminated DNA (Figure 3, *x*-axis: red) or suspected of being non-target DNA (Figure 3, *x*-axis: orange). This leaves *C. ecaudatus* represented for *Cytb*, 12S and the majority of 16S rRNA, whereas *C. yirratji* is represented only by a partial 16S rRNA sequence. Phylogenetic analyses



of the resulting 26-taxon MtNuc$_{26}$ data provide strong support for *Chaeropus* as sister to Peramelidae, to the exclusion of *Macrotis* (Figure 4A). Results are similar regardless of whether the rapidly evolving *Cytb* 3rd codon positions were RY-coded, with 1.00 BPP and 87–98% ML bootstrap support for both Peramelidae and the *Chaeropus*-Peramelidae grouping. Alternative placements for *Chaeropus* (including those in Figure 1) are strongly rejected by ML hypothesis testing (Table S5), except being sister to all extant peramelemorphians (*p* = 0.199). The primarily New Guinean bandicoots (*Peroryctes*, *Echymipera*, *Microperoryctes*) and the primarily Australian bandicoots (*Perameles*, *Isoodon*) were both strongly supported as monophyletic.

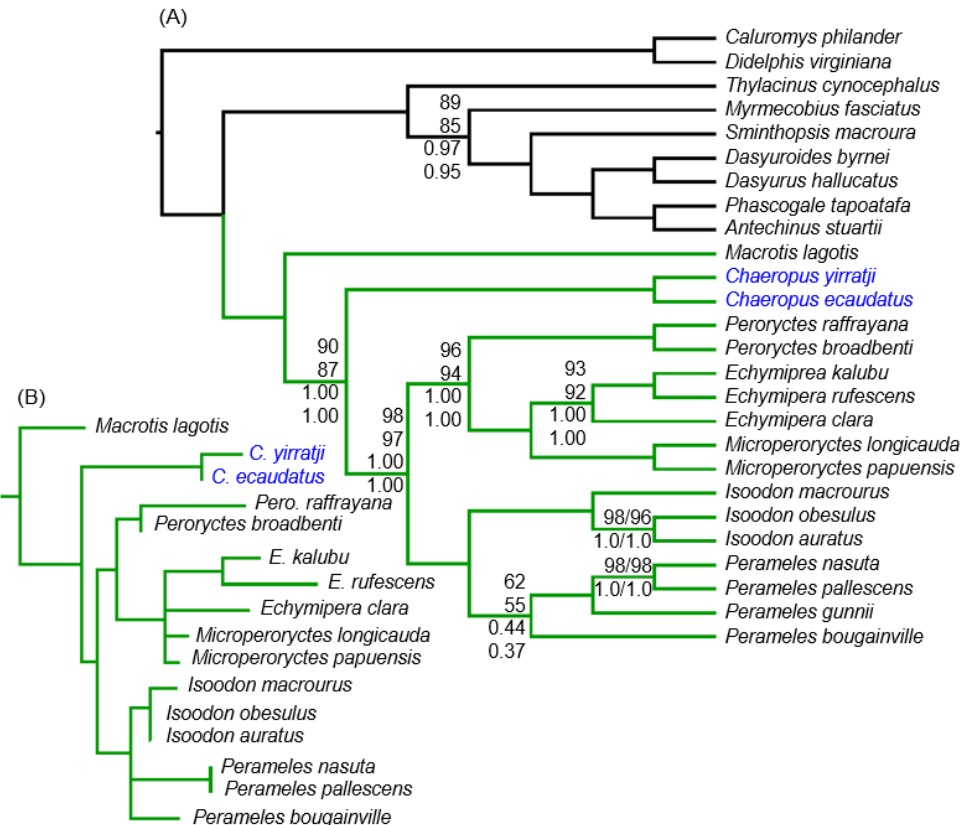

**Figure 4.** (**A**) MtNuc$_{26}$ maximum likelihood phylogeny focusing on Peramelemorphia (bandicoots and bilby, green branches) and without phylogenetic constraints. Clade support values at nodes, from top to bottom are ML-BP (NT data), ML-BP (RY-coded *Cytb* 3rd positions), BI-BPP (NT data), BI-BPP (RY-coded *Cytb* 3rd positions). Support values are not shown for clades when 100% for each measure. BP is ultrafast bootstrap in IQ-TRRE (-bb 10,000). BPP is Bayesian posterior probability in MrBayes. (**B**) IQ-TREE ML phylogram for the 361 bp segment of 16S rRNA that was deemed to be authentic (non-contaminated) for both *Chaeropus* species. The topology was constrained in agreement with the MtNuc$_{26}$ tree to ensure the appropriate phylogenetic context for inferring branch lengths with the truncated sequences.

Although the MtNuc$_{26}$ phylogenies help to clarify the genus-level placement of *Chaeropus*, the exclusion of putative non-target DNA and sites of uncertain homology in the alignment left the two pig-footed bandicoot species sharing only a 361 bp segment of 16S rRNA. To avoid branch length estimation biases associated with missing and non-overlapping sequences, we also inferred branch lengths on this short segment of the alignment. This resulted in the phylogeny shown in Figure 4B when constraining the topology to the MtNuc$_{26}$ tree (Figure 4A). *C. ecaudatus* has essentially zero branch-length for this 361 bp segment (Figure 4B).

### 3.3. Timescale of Chaeropus Evolution

The BEAST timetrees for the MtNuc$_{26}$ data with *Cytb* 3rd codon positions RY-coded (Figure 5) or treated as standard nucleotides (see Figure S4) have near-identical divergence times right across the marsupial tree. We will focus here on the RY-coded timetree. The median (and 95% HPD) estimate for *Chaeropus* diverging from their peramelid sister group is 16.8 (10.9–23.6) Mya. Peramelemorphia diverged from Dasyuromorphia at 55.9 (48.3–66.5) Mya, with the crown divergence of Peramelemorphia (bandicoots versus bilbies) at 22.6 (17.6–29.1) Mya. The two *Chaeropus* species were estimated to have diverged at 2.4 (0.3–6.1) Mya. That median estimate is roughly in the middle of the range of estimated divergences for bandicoot species pairs (Figure 5), and is most similar to the divergence of *Echymipera rufescens* versus *E. kalubu* at 2.38 (1.5–3.7) Mya. However, the divergence estimate between *C. yirratji* and *C. ecaudatus* is less precise, with the upper bound more than 20-fold older than the lower bound.

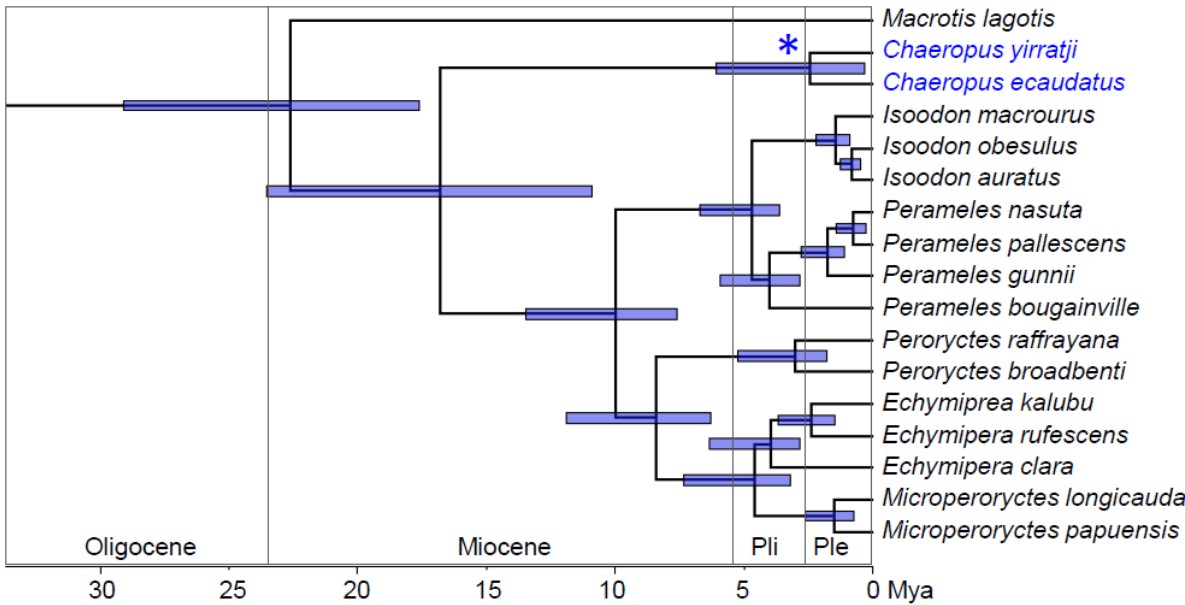

**Figure 5.** Molecular dated evolutionary timescale for peramelemorphian evolution, inferred in BEAST on the MtNuc$_{26}$ data matrix with *Cytb* 3rd codon positions RY coded. Blue bars show 95% highest posterior density for node ages. The asterisk indicates the approximate age of *Chaeropus baynesi* from the Fisherman's Cliff Local Fauna. Outgroup taxa and divergence times are provided in the Supplementary Information. Abbreviations: Pli; Pliocene, Ple; Pleistocene.

## 4. Discussion

### 4.1. MtDNA Authentication

Molecular systematics and evolutionary biology depend upon DNA sequences being authentic. Distinguishing target and non-target DNA with high probability is achievable by replicating sequences in different labs [23]. This is effectively achieved for *Chaeropus* only for a fragment of 16S rRNA (Figure 3B, green line), albeit with different species replicated. The absence of such replication for the other published *Chaeropus* sequences places the burden of evidence on phylogenetic methods. Here, we emphasize a cautious approach for accepting sequence authenticity, because contamination often does not manifest as 100% similarity to non-target taxa. This may be because GenBank does not cover the full diversity of potential non-target sequences or because contamination may be incorporated as a heterogenous mix of target and non-target DNA.

We employed a multi-pronged approach to identify non-target DNA fragments among published *Chaeropus* sequences. Initial sliding window BLASTs [46] provided fine precision, but this trades off against accuracy, due to being a similarity (phenetic) metric and using

narrow (150 bp) windows. BLAST similarity comparisons can be confounded by compositional bias and autapomorphy. We complemented the BLASTs with wider window (300 bp) MP bootstrap analyses on the densely sampled ($\text{Mt}_{192}$) backbone tree and by examining apomorphy distributions. Cumulative site likelihood (Figure 3) provided clear visualization of stark contamination examples, but ln*L* volatility associated with different substitution categories may obscure shorter or less evolutionarily distant non-target sequences.

Substitution rate variation between genes is an obstacle to circumscribing objective rules for identifying non-target DNA. For example, apparently authentic *Chaeropus* sequences among the slower evolving ribosomal RNA genes fit the expectation of sliding window MP affinities with Peramelemorphia, but faster evolving *Cytb* sequences often do not (Table S4). One solution was to run the sliding window MP bootstrap analyses for *Cytb* with the 3rd codon positions RY coded (Table S4), which removes the rapidly saturating transition signal, and has been shown to enhance deeper-level phylogenetic inference, including for marsupials [47]. The primary key to controlling for variation in substitution patterns within and between genes was to calibrate our BLAST and MP bootstrap expectations for *Chaeropus* by reference to a similarly evolutionarily distinct peramelephorian, the bilby (*Macrotis lagotis*), which has well-accepted modern sequences, (Tables S3 and S4).

RY coding *Cytb* 3rd codon positions and using *Macrotis* as a control provided a basis for deriving metrics that capture null expectations for *Chaeropus* sliding window BLAST and MP bootstrap results. Plotting BLAST identity ratio and highest anomalous MP bootstrap support metrics for each sliding window provided an expected distribution in which 95% of *Macrotis* and *C. ecaudatus* sliding windows clustered (Figure 2). With the exception of two outlier *Macrotis* windows that were explained by symplesiomorphy, other, more extreme sliding window outliers among *C. ecaudatus* and *C. yirratji* are best explained as non-target DNA.

Several of the published *C. yirratji* sequences include non-target DNA. 12S rRNA (AF131247) includes *Potorous* and *Macrotis* sequence (Figure 3A, Tables S3 and S4). Conversely, 16S rRNA (JF706364) was the first largely authentic DNA sequence published for *Chaeropus* (Westerman et al. [14]). This was a substantial achievement at that time. The specimen is relatively young (1901 CE); however, several leading ancient DNA labs failed to retrieve authentic DNA sequences from similarly preserved *Chaeropus* specimens, and Meredith et al. [13] noted the difficulty of amplifying *Chaeropus* DNA. The *C. ecaudatus* sequences are inferred to be mostly authentic, except for 16S rRNA (MK359295, MK359296), which includes a 392 bp *Isoodon macrourus* fragment (Table S3). This is surprising because these replicate sequences are identical (albeit from the same laboratory). However, this putative contamination may have originated in silico, since Travouillon et al. [6] used *I. macrourus* as the bioinformatic reference sequence.

The *C. yirratji Cytb* (JF718363) sequence provides a complex example of contamination. The sequence BLASTs with the marsupial order Dasyuromorphia, closest to two dunnarts (*Sminthopsis*). Closer inspection revealed that the sequence is a chimera of marsupial DNA and a fragment of avian DNA that also contaminates the two dunnart sequences. That avian fragment in JF718363 and in both dunnarts matches (129/130 bp) to several crested bellbird (*Oreoica gutturalis*) sequences. This avian contamination explains both *Chaeropus* nesting within *Sminthopsis*, based on *Cytb* in Upham et al. [19], and the anomalous *Sminthopsis Cytb* phylogeny noted by Krajewski et al. [48]. Recognizing non-authentic DNA on GenBank remains a vexed issue. However, we recently proposed a mechanism for updating taxonomic attributions [49] that gives original contributors the first option for revision, and may also assist with flagging or correcting non-authentic sequences.

### 4.2. Peramelemorphian Systematics

Non-target DNA incorporated in published *Chaeropus* sequences has further blurred the affinities of pig-footed bandicoots and inflated estimates of the temporal divergence between *C. ecaudatus* and *C. yirratji*. Phylogenetic analysis of the combined mitochondrial and nuclear dataset ($\text{MtNuc}_{26}$) after excluding the likely and suspected non-target DNA

fragments brings *C. ecaudatus* and *C. yirratji* into closely divergent monophyly. In turn, *Chaeropus* is supported as sister to Peramelidae (Figure 4A). The alternative affinities of *Chaeropus* found in previous studies (Figure 1) will have been substantially influenced by the inclusion of non-target DNA fragments. All of the *Chaeropus* DNA used here is mitochondrial. Good agreement between mtDNA and nuclear DNA for similar magnitudes of statistical and branch length support in other marsupial studies lends confidence to the present result being robust to incomplete lineage sorting (see [33,34,42,50]). However, deep mitochondrial introgression needs to be ruled out. Although rare, the swamp wallaby (*Wallabia bicolor*) provides a cautionary example, with its mtDNA captured from a now extinct, deeper diverging kangaroo [37,38]. Hence, nuclear data will be required to confirm the placement of *Chaeropus* as sister to Peramelidae.

The sister relationship of *Chaeropus* to extant peramelids, and inferred temporal divergence between these two clades of 16.8 (10.9–23.6) Mya (Figure 5) are equivocal for assigning pig-footed bandicoots to their own family. Several other marsupial family crown ages may be older, such as Dasyuridae, Acrobatidae and Burramyidae (see [33,34]). However, *Chaeropus* morphology alludes to functional and ecological distinctiveness that sets them apart from both extant peramelemorphian families, Peramelidae and Thylacomyidae, and in our view justifies Groves [51] placing *Chaeropus* in its own family, Chaeropodidae.

Deeper in the tree, the estimate for the crown Peramelemorphia (bandicoots versus bilby) divergence of 22.6 (17.6–29.1) Mya accords with the earliest relatively well-supported crown fossil taxon being the ~14 Mya thylacomyid, *Liyamayi dayi* [52]. Unfortunately, the absence of tight calibration limits molecular dating precision. The 95% credible interval for the peramelemorphian crown divergence is also consistent with the more tentative assignment of the 24.9 Mya *Bulbadon warburtonae* to Thylacomyidae [20]. However, Travouillon et al.'s [20] matrix-based phylogenetic analysis placed *B. warburtonae* in a polytomy that leaves its crown affinity unresolved. Further investigation into the affinities of *B. warburtonae* and other Late Oligocene peramelemorphians, ideally with more complete material, will be important for clarifying basal bandicoot (and bilby) relationships and for more precisely calibrating molecular or total evidence dating.

*4.3. Chaeropus Evolution*

Only one fossil chaeropodid has been published. The 2.92–2.47 Mya *Chaeropus baynesi* from the Fisherman's Cliff Local Fauna is known from several molars that showcase the transition towards increased herbivory, but not yet grazing [21]. Although dental microwear analysis has not yet been undertaken, Travouillon [21] concluded from gross molar morphology that grazing specialization in *Chaeropus* was more recent, and was probably a response to Pleistocene drying. Travouillon et al.'s [6] subsequent molecular dates appear out of step with this scenario, instead implying far earlier grazing, with the more specialized *C. ecaudatus* and *C. yirratji* diverging at 8.6 (95% CI: 3.2–13.4) Mya. This temporal anomaly is resolved in the present study by excluding the non-target DNA, whereby the divergence between the modern species falls to 2.4 (0.3–6.1) Mya (Figure 5). Our date may even be overestimated if branch length asymmetry on the tree inferred from the 361 bp fragment that is shared by both *Chaeropus* species hints at the *C. yirratji* Sanger sequence retaining some errors compared to the zero-branch length *C. ecaudatus* (Figure 4B). However, such speculation is premature, since the same tree shows similar branch length asymmetry among other bandicoot species pairs. Nevertheless, our more recent divergence timing for *Chaeropus* is consistent with grassland expansion [53] and increased grazing adaptation among kangaroos [32,54] from 6 Mya or younger. The shallow divergence between *C. ecaudatus* and *C. yirratji* (Figure 5) does lessen the molecular case for their species-level distinctiveness, but is not inconsistent with Travouillon et al.'s [6] taxonomy. Indeed, with so little DNA contributing to *C. yirratji*, it is prudent to prioritise the morphological arguments for recognising both *C. ecaudatus* and *C. yirratji*.

All crown bandicoots and bilbies had ancestors that were semi-fossorial omnivores [34,55] and it is from this ancestry that *Chaeropus* evolved an array of appendicular, dental and

digestive system traits indicative of cursorial and grazing behaviours (see [6]). The extent to which *Chaeropus* retained an insectivorous component to its diet is clouded by conflicting evidence. Examination of several faecal pellets found only grass [5,56]. Conversely, some Aboriginal observations attributed ant and termite feeding to *Chaeropus* [57], while theoretical considerations place their ~200–500 g mass below size thresholds for exclusive grazing [58,59]. The pointed snout is also suggestive of insectivory, even soil/sand probing as in other bandicoots. A relevant question here, is whether this reflects the habits of pig-footed bandicoots or evolutionary inertia— is this a ghost of their recent ecological past?

Plio-Pleistocene evolution of predominantly grazing bandicoots would be remarkable as a possible incursion across foraging ecospaces that map to marsupial orders (Figure 6A) and may have been phylogentically conserved for 50 million years. *Chaeropus* may have evolved into ecospace occupied by the order Diprotodontia, particularly macropodoids (kangaroos and bettongs). Indeed, *Chaeropus* might be the only example among recent marsupial fauna, of an incursion across an evolutionarily stable niche discontinuity (ESND, see [60]). An earlier broad-scale foraging ecospace overlap between orders involved dasyuromorphians and thylacoleonids (marsupial lions) [61].

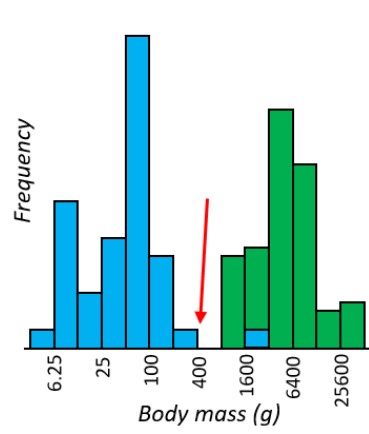

**Figure 6.** (**A**) Foraging ecospace distribution among extant Australian marsupials, with genera denoted as circles for the four orders, Diprotodontia (green), Peramelemorphia (orange), Notoryctemorphia (brown) and Dasyuromorphia (red). Diet axis: Herb (herbivorous), Om$_S$ (plant-specialized omnivorous), OmG (generalized omnivorous), Anim (Animalivory). See Table S6 for definitions and scoring. Foraging height axis: Fos (fossorial), SF (semi-fossorial), Ter (terrestrial), MT (mostly terrestrial), Sc (scansorial) and Arb (arboreal). Dashed arrows indicate four alternative foraging ecospace transitions along the lineage leading to *Chaeropus*. (**B**) Frequency distribution average adult body mass (g) among all extant hopping mammals, which include rodents (blue) and macropodoids (green). The red arrow indicates estimated body mass for *Chaeropus*.

Three of the four possible foraging ecospace placements for *Chaeropus* in Figure 6A (into green patches) would represent an incursion across the ESND between bandicoots and diprotodontians, albeit into new (grassland) ecospace. Two aspects of the biology of *Chaeropus* and potential competition with macropods may be relevant: (1) In the ancestors of pig-footed bandicoots, small size near the energetic feasibility limit for a grazer may shift the balance of selection pressure in favour of gracile, ungulate-like legs for energetically efficient cursorial locomotion over selection for mechanical advantage in digging. The smallest predominantly grass-feeding macropods (see [62]), such as the rufus hare-wallaby (*Lagorchestes hirsutus*) are larger, averaging ~1.3 kg. (2) The mass of the quadrupedal *Chaeropus* falls in the trough of the binomial body mass distribution for mammalian hoppers (Figure 6B). Hopping appears to be most advantageous for predator avoidance at small body sizes [63] and imparts energetic advantages principally at larger sizes [64]. Moreover, macropod locomotion tends to be inefficient at low speeds [65]. Thus, an ecological

opportunity for a small grazer feeding more or less constantly across the landscape rather than between more distant patches might be more accessible for a bandicoot evolving specialized quadrupedal locomotion than for further shrinking a macropod.

*Chaeropus* evolution may have instead not violated the conservation of ESNDs between marsupial orders—if invertebrates remained an important component of their diet, even seasonally, and if they foraged terrestrially as their limb morphology may imply [6], then *Chaeropus* evolved into an ecospace that is largely unoccupied by other modern marsupials (Figure 6A, open patch). The paucity of marsupials and dominance of rodents now occupying Australia's mammalian terrestrial omnivore foraging ecospace alludes to the potential importance of looking beyond the intrinsic biology of pig-footed bandicoots to understand their evolution. In particular, the temporal coincidence of the *Chaeropus* ecological shift revealed by the transitional *C. baynesi* and the Pliocene diversification of murid rodents begs the question of how the newly arrived placental omnivores shifted the balance of competition among marsupials for ecospace occupation. This broader view will be critically informed by further fossil evidence. However, genomics may offer valuable insights for testing the alternative ecological transition pathways for *Chaeropus*, by identifying functional mutations, such as in chitinase genes, which have marked dietary transitions from omnivory to more exclusive herbivory among placental mammals [66].

**Supplementary Materials:** The following supporting information can be downloaded at: https://www.mdpi.com/article/10.3390/d14050352/s1, Figures S1–S4: Supplementary figures.pdf; Tables S1–S7: Supplementary tables.pdf.

**Author Contributions:** Conceptualization, M.J.P.; methodology, M.J.P.; formal analysis, M.J.P. and M.C. (Manuela Cascini); investigation, M.J.P., M.C. (Manuela Cascini) and M.C. (Mélina Celik); data curation, M.J.P., M.C. (Manuela Cascini) and M.C. (Mélina Celik); writing—original draft preparation, M.J.P.; writing—review and editing, M.J.P., M.C. (Manuela Cascini) and M.C. (Mélina Celik); visualization, M.J.P. and M.C. (Mélina Celik); supervision, M.J.P.; funding acquisition, M.J.P. All authors have read and agreed to the published version of the manuscript.

**Funding:** This research was funded by The Australian Research Council, grant number "DP190103636".

**Institutional Review Board Statement:** Not applicable.

**Informed Consent Statement:** Not applicable.

**Data Availability Statement:** Body mass and foraging ecospace data are provided in the Supplementary information. Original DNA sequences are available at GenBank and alignments are available upon request.

**Acknowledgments:** We thank Kenny Travouillon, Robin Beck and three anonymous reviewers for valuable comments on the manuscript.

**Conflicts of Interest:** The authors declare no conflict of interest. The funders had no role in the design of the study; in the collection, analyses, or interpretation of data; in the writing of the manuscript, or in the decision to publish the results.

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
