# Peer review of "Identifying Complex DNA Contamination in Pig-Footed Bandicoots Helps to Clarify an Anomalous Ecological Transition"

_diversity, doi:10.3390/d14050352_

Round 1

Reviewer 1 Report

Phillips and co-authors thoroughly examine the published mitochondrial DNA sequences of the extinct pig-footed bandicoots, searching for possible contamination which can explain the very variable phylogenetic affinities which have previously been published for the genus.
The manuscript is clearly written and the conclusions sound. The methods employed here to tease out regions of DNA sequences which may be afflicted by contamination or errors are of interest to an audience beyond those specifically interested in Chaeropus evolution.

I have only a few minor comments to make.

Pg 2 line 43 Change 'It employed' to 'They employed'

Figure 2
Is there any way to also code the spots for gene as well as species? Or does that just get too messy?

It would be good to note in Fig 3 that as MK369295 and MK369296 are identical this applies equally to both sequences.

Pg 10 3rd paragraph. 
What are the chances the errors in the first third of C. yirratji 16S rRNA (JF706364) are caused by poor electropherogram cleanup (ie messy sequence that was not called properly or  trimmed) rather than by contamination?

Pg 13 Start of discussion. You mention it later on but it is relevant here - even sequencing in separate labs does not guarantee the published results are right just be cause they look the same. It is still possible to introduce 'in silico' contamination here with reference sequences mistakenly getting included in the consensus.

Pg 16 Figure 6
The y axis abbreviations in fig 6A appear to be obscured, I can only see Foss. Also the caption could explain the abbreviations on the x axis as well (Herb, Oms, Omg, Anim).

What is the source of the foraging ecospace information in fig 6a/Table S6? I can only see the body mass PanTHERA dataset reference.

Do the authors have any comment on how we deal with the increasing number of sequences with significant errors on Genbank? How will future studies that download Chaeropus sequences from Genbank to include in their research know to avoid the ones you list here, besides a thorough literature search? Obviously this is a bigger issue than the present study.

Reviewer 2 Report

In the manuscript “Indentifying Complex DNA Contamination in Pig-footed Bandicoots Helps to Clarify an Anomalous Ecological Transition” Phillips et al. identified contamination in publically available sequences of the genus Chaeropus and used the filtered dataset for phylogenetic analyses.

For me the manuscript is written in an unnecessary complex form that is honestly very difficult to read and overall it is just boring.

Important parts are not properly and understandably discussed. With no word it is mentioned what type of data the published sequences are derived from.  I assume as it is most common for single genes that it is Sanger sequencing. It is not discussed how it is possible that apparently sections of one Sanger read are contamination while other parts are not. It is also not stated how long the published sequences even are or how many windows have been generated.

Also as a side-note the time for single or few gene phylogenies even for extinct species should be over. It is quite easy even from 100-250 year old mammalian/marsupial museum samples to generate whole genome sequences or at least complete mitochondrial sequences, which would be much more informative.

I am sorry that I cannot give a more positive review of this manuscript but as it is right now I would not publish it.

Reviewer 3 Report

please see attached my comments.

Reviewer 4 Report

Phillips et al. present a timely analysis of historical DNA sequences generated from museum specimens of an enigmatic Australian marsupial. Their results confirm a long-held suspicion of many marsupial and aDNA genetics researchers in Australia that many/most of Mike Westerman's DNA sequences on GenBank are chimeric and/or contain errors. It's disappointing that sequences from Kenny Travouillon's more recent paper also appear to be chimeric - I don't have access to Zootaxa so can't see how they generated the 16S rRNA sequence. But if the suggestion of bioinformatic mis-mapping of Illumina reads is correct, then it suggests that someone should obtain all the raw data and remap those short reads more carefully.

The authors need to do a very careful spell check of the manuscript - I noted a number of obvious spelling mistakes that should show up in a Word spell/grammar check.

Several of the Genbank accession numbers cited in the manuscript are also incorrect - somewhat embarassing to be complaining about chimeric DNA sequences in other's work, but then get the GenBank accession numbers wrong! For example MK369295 (cited multiple times in the paper) takes me to a GenBank entry for Listeria not pig-footed bandicoot.

Round 2

Reviewer 2 Report

The authors explained some of my concerns in their reply, and maybe I was a bit too critical in my first assessment. I am happy with the improvements by the authors, yet, I still think it could be written in a way that is easier to understand.